# Causal Inference using LLM-Guided Discovery

**Aniket Vashishtha[1], Abbavaram Gowtham Reddy[2], Abhinav Kumar[3],**
**Saketh Bachu[2], Vineeth N Balasubramanian[2], Amit Sharma[1]**

[1]Microsoft Research, India
[2]IIT Hyderabad, India
[3]Massachusetts Institute of Technology, USA
{t-aniketva, amshar}@microsoft.com,
{cs19resch11002, vineethnb, saketh.bachu}@cse.iith.ac.in, akumar03@mit.edu

## Abstract

At the core of causal inference lies the challenge of recovering the underlying causal graphs based on observational data. Recent work aggregates the results of edge-wise prompts to infer the full graph structure using Large Language Models (LLMs). However, graph structure cannot be identified uniquely using such localized prompts for each edge since the existence of an edge depends on which other nodes are included in the node set. Therefore, we propose a simpler property of the causal graph, the *topological order*, that can be estimated reliably using localized LLM prompts. Moreover, for downstream tasks like effect inference, the topological order is sufficient to identify causal effect. Hence we LLMs as virtual domain experts and propose a novel localized prompt based on triplets to infer the causal order. Acknowledging LLMs' limitations, we also study possible techniques to integrate LLMs with established causal discovery algorithms, including constraint-based and score-based methods. Extensive experiments demonstrate that our approach significantly improves causal ordering accuracy as compared to discovery algorithms, and in turn decreases the error of downstream effect estimation algorithms.

## 1 Introduction

A key question for studies across scientific fields such as epidemiology, economics, and atmospheric sciences is estimating the *causal effects* of variables on an outcome variable. Inferring causal effects from observational data, however, is a difficult task because the effect estimate depends on the causal graph considered in the analysis. While there has been progress in graph discovery algorithms, especially for specific parametric settings (Shimizu et al. 2006; Hoyer et al. 2008; Hyvärinen et al. 2010; Rolland et al. 2022), studies on real-world datasets such as from atmospheric science and healthcare (Huang et al. 2021; Tu et al. 2019) show that inferring the causal graph from data remains a challenging problem in practice (Reisach, Seiler, and Weichwald 2021). Therefore, causal effect inference studies often rely on a human expert to provide the causal graph.

In this paper, based on the fact that the *topological causal order* over the graph variables is enough for effect inference (see Proposition 4.1), we leverage LLMs as virtual domain experts to propose an automated method to obtain causal order (and hence causal effect). Moreover, providing the *order*

between variables is the right question to ask experts because it depends only on the variables under question, unlike the existence of a graph *edge* that depends on which other variables are present (to account for direct and indirect effects). For example, consider the data-generating process, *lung cancer → doctor visit → positive Xray*. If asked, an expert would affirm a causal edge from *lung cancer* to *positive Xray* (indeed, such an edge exists in the BNLearn *Cancer* dataset (Scutari and Denis 2014)). However, if they are told that the set of observed variables additionally includes *doctor visit*, then the correct answer would be to not create a direct edge between lung cancer and positive Xray, but rather create edges mediated through *doctor visit*. Note that the causal order, *lung cancer ≺ positive Xray* remains invariant in both settings.

This observation has implications on using LLMs for inferring graph structure, where existing work prompts the LLM about each causal edge separately (Kıcıman et al. 2023; Long, Schuster, and Piché 2022). As we argued above, accuracy of such pairwise prompting is not reliable since it depends critically on which other nodes are included in the node set. The simple solution of constructing a prompt that includes all other nodes may be infeasible, especially for large graphs. As a result, we argue that graph structure may not be a suitable output to expect from LLMs (or any other expert). Instead, our main insight is that a simpler graph property, the causal order, can be inferred locally and is not affected by availability of other nodes. Moreover, for downstream tasks like effect inference, the causal order is sufficient to identify the causal effect; full graph structure is not necessary. Our empirical results on six benchmark datasets show that LLMs like GPT-3.5 and GPT-4 can approximate experts' causal order capabilities. To do so, we propose a novel triplet-based prompting strategy that performs better than pairwise prompts (Kıcıman et al. 2023; Willig et al. 2022; Long, Schuster, and Piché 2022) for determining the causal order. The triplet prompt also produces significantly lesser cycles in the output graph.

Still, LLMs can exhibit unknown failure modes. Therefore, we propose two algorithms to combine existing graph discovery algorithms with LLMs: one employs LLM causal order to guide a constraint-based algorithm (e.g., PC) in orienting undirected edges, while the second incorporates LLM causal order as a prior for a score-based algorithm like CaMML. We find that LLM-enhanced algorithms outperform base causal discovery methods in inferring causal order. The method-

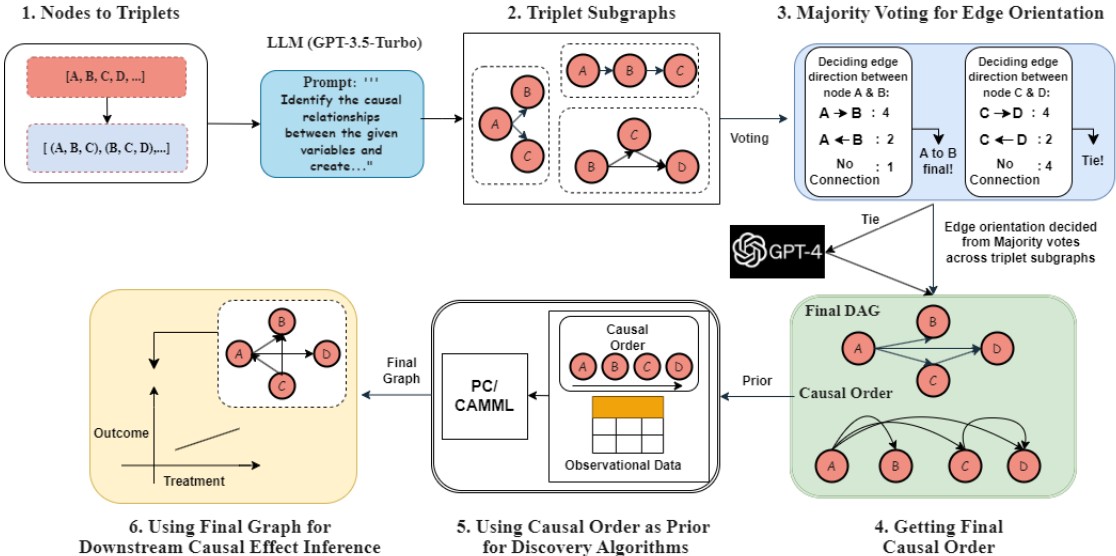

Figure 1: The *LLM-augmented* causal inference process based on inferring causal order. We propose a triplet-based prompting technique to infer all three-variable subgraphs and aggregate them using majority voting to produce a causal order. The causal order (optionally combined with discovery algorithms like PC or CaMML) can then be used to identify a valid back-door adjustment set. Ties in causal order are broken using GPT-4.

ology is illustrated in Figure 1. Our contributions include,

- We propose prompting strategies for estimating the order using LLMs, which is more reliable than estimating the graph structure using LLMs.
- We propose algorithms for inferring causal order using a novel triplet prompting strategy, using only LLMs and combining them with existing discovery algorithms.

## 2 Related Work

Historically, causal discovery (Glymour, Zhang, and Spirtes 2019; Rolland et al. 2022; Teyssier and Koller 2005; Zheng et al. 2018; Lachapelle et al. 2020) and causal effect inference (Pearl 2009) have been studied separately. Instead of using the learned graph for effect inference (Hoyer et al. 2008; Mooij et al. 2016; Maathuis et al. 2010; Gupta, Childers, and Lipton 2022), we demonstrate a simpler combination of approaches: causal order suffices instead of the entire graph.

Our work is related to LLM-based knowledge-driven causal discovery (Kıcıman et al. 2023; Ban et al. 2023; Long, Schuster, and Piché 2022; Willig et al. 2022). Unlike causal discovery algorithms that use statistical patterns in the data, LLM-based algorithms use metadata such as variable names. These methods use LLMs to predict causal structure over a set of variables by aggregating the results of an edge-wise prompt for each pair of variables (Kıcıman et al. 2023; Long et al. 2023; Willig et al. 2022; Long, Schuster, and Piché 2022). Instead, we show that the existence of an edge may not be identified if we do not know other existing variables; hence causal order is a more suitable output to elicit from LLMs. We also propose a triplet-based prompt for inferring the causal order, which may be of independent interest in prompting LLMs for causality.

Since LLMs may exhibit errors, a more principled approach may be to combine LLMs with existing discovery algorithms. (Long et al. 2023) use LLM to improve the output of a constraint-based algorithm for full graph discovery and (Ban et al. 2023) use LLMs as priors for existing score-based causal discovery methods. We extend this idea to causal order estimation by proposing LLM-adaptations for constraint- and score-based algorithms.

## 3 Background and Problem Formulation

Let $\mathcal{G}(\mathbf{X}, \mathbf{E})$ be a causal directed acyclic graph (DAG) consisting of a set of variables $\mathbf{X} = \{X_1, \dots, X_n\}$ and a set of directed edges $\mathbf{E}$ among the variables in $\mathbf{X}$. A directed edge $X_i \rightarrow X_j \in \mathbf{E}$ denotes the *direct* causal influence of the variable $X_i$ on the variable $X_j$. Let $pa(X_i) = \{X_k | X_k \rightarrow X_i\}$, $de(X_i) = \{X_k | X_k \leftarrow \cdots \leftarrow X_i\}$ denote the set of *parents* and *descendants* of $X_i$ respectively. A sequence $\pi$ of variables $\mathbf{X}$ is said to be a topological order iff for each edge $X_i \rightarrow X_j \in \mathbf{E}$, $\pi_i < \pi_j$.

We focus on a downstream application of causal graph discovery called causal effect inference. The average causal effect (ACE) of a variable $X_i$ on a variable $X_j$ is defined as $ACE_{X_i}^{X_j} = \mathbb{E}[X_j | do(X_i = x_i)] - \mathbb{E}[X_j | do(X_i = x_i^*)]$ (Pearl 2009). Here, $X_i$ is called the *treatment* variable and $X_j$ is called the *target* variable. $do(X_i = x_i)$ denotes an external intervention to the variable $X_i$ with the value $x_i$. The interventional quantity $\mathbb{E}[X_j | do(X_i = x_i)]$ is different from conditional $\mathbb{E}[X_j | X_i = x_i]$ since it involves setting the value of $X_i$ rather than conditioning on it. To estimate the quantity $\mathbb{E}[X_j | do(X_i = x_i)]$ from observational data, the back-door adjustment formula (Pearl 2009) is used. Given a DAG $\mathcal{G}$, a set of variables $\mathbf{Z}$ satisfies back-door criterion relative to a pair of treatment and

target variables $(X_i, X_j)$ if (i) no variable in **Z** is a descendant of $X_i$; and (ii) **Z** blocks every path between $X_i$ and $X_j$ that contains an arrow into $X_i$. Where a *path* in a causal DAG is a sequence of unique vertices $X_i, X_{i+1}, \ldots, X_j$ with a directed edge between each consecutive vertices $X_k$ and $X_{k+1}$ (either $X_k \to X_{k+1}$ or $X_{k+1} \to X_k$). If a set of variables **Z** satisfies the back-door criterion relative to $(X_i, X_j)$, $\mathbb{E}[X_j | do(X_i = x_i)]$ can be computed using the formula: $\mathbb{E}[X_j | do(X_i = x_i)] = \mathbb{E}_{\mathbf{z} \sim \mathbf{Z}} \mathbb{E}[X_j | X_i = x_i, \mathbf{Z} = \mathbf{z}]$ (Theorem 3.3.2 of Pearl (2009)). To ensure causal effect identifiability, we make the no-latent confounding assumption.

## 4 Causal Order Suffices for Effect Estimation

We start with the fact in Proposition 4.1 that the causal order is sufficient to find a valid back-door set and discuss why causal order is easier to elicit from experts than the DAG.

### 4.1 Causal Order Provides a Valid Back-Door Set

**Proposition 4.1.** *(Pearl 2009; Cinelli, Forney, and Pearl 2022a) Under the no-latent confounding assumption, given a pair of treatment and target variables $(X_i, X_j)$, $\mathbf{Z} = \{X_k | \pi_k < \pi_i\}$ is a valid adjustment set relative to $(X_i, X_j)$ for any topological order $\pi$ of nodes $X_i, \ldots, X_n$.*

Proofs of all propositions are in Appendix § A. Propn 4.1 states that all the variables that precede the treatment variable in a topological order $\pi$ of $\mathcal{G}$ constitute a valid adjustment set. Note that the set **Z** may contain variables that are not necessary to adjust for, e.g., ancestors of only treatment or only target variables. For statistical estimation, ancestors of target variable are beneficial for precision whereas ancestors of treatment can be harmful (Cinelli, Forney, and Pearl 2022b). On balance though, causal effect practitioners tend to include all confounders that do not violate the back-door criterion; we are following the same principle. In practice, however, we may not know the true order. To evaluate the goodness of a given causal order, we use the topological divergence metric from (Rolland et al. 2022) (for an example, see Fig. 3). The topological divergence of an estimated topological order $\hat{\pi}$ with ground truth adjacency matrix $A$, denoted by $D_{top}(\hat{\pi}, A)$, is defined as $D_{top}(\hat{\pi}, A) = \sum_{i=1}^{n} \sum_{j: \hat{\pi}_i > \hat{\pi}_j} A_{ij}$.

Where $A_{ij} = 1$ if there is a directed arrow from node $i$ to $j$ else $A_{ij} = 0$. $D_{top}(\hat{\pi}, A)$ counts the number of edges that cannot be recovered due to the estimated topological order $\hat{\pi}$.

### 4.2 $D_{top}$ Is a Valid Metric for Effect Estimation

Below, we show that $D_{top}$ is a valid metric to check the correctness of the estimated causal effects. That is $D_{top}(\hat{\pi}, A)$ being 0 is equivalent to obtaining the correct back-door adjustment set from $\hat{\pi}$ (Proposition 4.1).

**Proposition 4.2.** *For an estimated topological order $\hat{\pi}$ and a true topological order $\pi$ of a causal DAG $\mathcal{G}$ with the corresponding adjacency matrix $A$, $D_{top}(\hat{\pi}, A) = 0$ iff $\mathbf{Z} = \{X_k | \hat{\pi}_k < \hat{\pi}_i\}$ is a valid adjustment set relative to $(X_i, X_j)$, $\forall \pi_i < \pi_j$.*

We now compare $D_{top}$ to structural hamming distance (SHD), a common metric for evaluating graph discovery

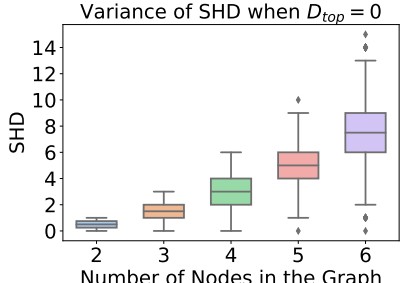

Figure 2: Variability of SHD with consistent $D_{top} = 0$.

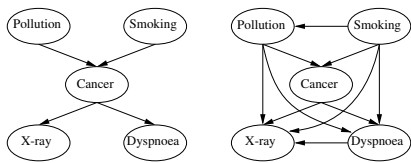

Figure 3: **Left:** Causal graph of Cancer dataset. **Right:** GPT-3.5's estimated causal graph. GPT-3.5 gets causal order correct at the cost of higher SHD score Here $D_{top} = 0$ and $SHD = 6$

algorithms. Given a true causal DAG $\mathcal{G}$ and an estimated causal DAG $\hat{\mathcal{G}}$, SHD counts the number of missing, falsely detected, and falsely directed edges in $\hat{\mathcal{G}}$. We note that SHD can be very high even when $D_{top} = 0$ and a valid back-door set can still be inferred. This result is of significance since most estimated graphs (included those that are LLM-generated (Ban et al. 2023; Long et al. 2023)) are evaluated on SHD.

**Proposition 4.3.** *In a causal DAG $\mathcal{G}$ with $N$ levels in the level-ordering of variables where the level $i$ contains $n_i$ variables, $\exists \hat{\mathcal{G}}$ s.t. $SHD(\hat{\mathcal{G}}, \mathcal{G}) \geq \sum_{i=1}^{N-1} \left( n_i \times \sum_{j=i+1}^{N} n_j \right) - |\mathbf{E}|$ and $D_{top}(\hat{\pi}, A) = 0 \, \forall \hat{\pi}$ of $\hat{\mathcal{G}}$.*

where a level order refers to a systematic assignment of levels to variables. This assignment begins with the set of variables $\{X_i | pa(X_i) = \emptyset\}$ at level 0. Subsequently, each of the remaining variables is assigned a level $i$ such that all nodes within a given level $i$ has a directed path of length $i$ from one/more nodes in level 0. Figure 2 shows the limitations of SHD empirically. Given a fixed number of nodes, we sample a graph at random as the 'ground-truth' and then consider all graph orientations of the same size (number of nodes) such that $D_{top} = 0$ with respect to ground-truth graph. For this set of graphs, we compute the SHD with respect to the ground-truth graph. Notice that SHD exhibits high variance. For graphs with six nodes, SHD can vary from 0 to 14 even as $D_{top} = 0$ and back-door set validity stays the same. Fig. 3 shows this phenomenon on a real-world BNLearn dataset, *Cancer*. The estimated graph (right panel) has $D_{top} = 0$ with respect to the true graph (left) and yields valid back-door identification sets. However, its SHD is high (6), showing the disconnect between SHD and causal effect identification.

| | Dataset | PC | SCORE | ICA LINGAM | Direct LINGAM | NOTEARS | CaMML | Ours (PC+LLM) | Ours (CaMML+LLM) |
|---|---|---|---|---|---|---|---|---|---|
| **N = 250** | Earthquake | 0.16±0.28 | 4.00±0.00 | 3.20±0.39 | 3.00±0.00 | 1.80±0.74 | 2.00±0.00 | **0.00±0.00** | **0.00±0.00** |
| | Cancer | **0.00±0.00** | 3.00±0.00 | 4.00±0.00 | 3.60±0.48 | 2.00±0.00 | 2.00±0.00 | **0.00±0.00** | **0.00±0.00** |
| | Survey | 0.50±0.00 | 5.00±0.00 | 6.00±0.00 | 6.00±0.00 | 3.20±0.39 | 3.33±0.94 | **0.00±0.00** | 1.00±0.21 |
| | Asia | 2.00±0.59 | 5.00±0.00 | 6.20±0.74 | 7.00±0.00 | 4.00±0.00 | 1.85±0.58 | 0.00±1.00 | **0.00±0.00** |
| | Asia-M | 1.50±0.00 | 5.00±0.00 | 7.60±0.48 | 6.20±1.16 | 3.40±0.48 | **1.00±0.00** | **1.00±0.00** | 1.21±0.30 |
| | Child | 5.75±0.00 | 8.80±2.70 | 12.8±0.97 | 13.0±0.63 | 15.0±1.09 | **3.00±0.00** | 4.00±0.00 | 3.53±0.45 |
| | Neuropathic | 4.00±0.00 | 6.00±0.00 | 13.0±6.16 | 10.0±0.00 | 9.00±0.00 | 10.4±1.95 | **3.00±0.00** | 5.00±0.00 |
| **N = 500** | Earthquake | 0.75±0.25 | 4.00±0.00 | 3.20±0.39 | 3.40±0.48 | 1.20±0.40 | **0.00±0.00** | **0.00±0.00** | **0.00±0.00** |
| | Cancer | **0.16±0.28** | 3.00±0.00 | 3.40±0.48 | 3.00±0.00 | 2.00±0.00 | 1.00±0.00 | 0.33±0.57 | 1.00±0.00 |
| | Survey | 1.25±0.00 | 4.00±0.00 | 6.00±0.0 | 6.00±0.00 | 3.40±0.48 | 3.39±0.08 | **1.00±0.00** | 1.00±0.21 |
| | Asia | 3.06±0.00 | 5.00±0.00 | 5.60±0.48 | 7.00±0.00 | 3.20±0.39 | 3.81±0.39 | 1.00±0.00 | **0.97±0.62** |
| | Asia-M | 2.00±0.00 | 6.00±0.00 | 7.60±0.48 | 5.00±0.00 | 3.80±0.39 | 2.00±0.00 | 1.00±0.00 | **0.17±0.45** |
| | Child | 8.09±1.17 | 6.20±1.32 | 12.2±0.74 | 10.6±1.35 | 15.4±0.48 | **2.00±0.00** | 5.00±1.73 | 2.00±0.00 |
| | Neuropathic | 7.50±0.00 | 6.00±0.00 | 9.00±1.41 | 13.0±0.00 | 11.0±0.00 | 5.32±0.57 | 8.00±0.00 | **3.56±0.73** |
| **N = 1000** | Earthquake | 1.33±0.57 | 4.00±0.00 | 3.00±0.00 | 3.00±0.00 | 1.00±0.00 | **0.00±0.00** | 0.66±0.57 | **0.00±0.00** |
| | Cancer | 1.33±0.57 | 3.00±0.00 | 3.00±0.00 | 3.00±0.00 | 2.00±0.00 | 1.60±0.48 | 1.33±0.57 | **0.00±0.00** |
| | Survey | **1.00±0.00** | 4.00±0.00 | 5.80±0.39 | 5.40±0.48 | 3.20±0.39 | 2.71±0.27 | **1.00±0.00** | 1.00±0.00 |
| | Asia | 5.00±0.00 | 4.00±0.00 | 6.20±0.74 | 6.60±0.48 | 3.40±0.48 | 1.75±0.43 | 5.00±0.00 | **0.97±0.62** |
| | Asia-M | 1.50±0.00 | 4.00±0.00 | 8.00±0.00 | 5.20±0.39 | 3.40±0.48 | 2.04±0.51 | 1.00±0.00 | **0.65±0.47** |
| | Child | 8.25±0.00 | 3.80±0.74 | 12.2±1.72 | 11.8±0.74 | 15.2±0.97 | **2.00±0.00** | 7.0±0.00 | 2.00±0.40 |
| | Neuropathic | - | 6.00±0.00 | 4.00±0.81 | 12.0±0.00 | 12.0±0.00 | 5.54±0.75 | - | **2.71±1.69** |
| **N = 5000** | Earthquake | 0.50±0.86 | 4.00±0.00 | 2.80±0.39 | 3.00±0.00 | 1.00±0.00 | 0.80±0.97 | **0.00±0.00** | **0.00±0.00** |
| | Cancer | 1.33±0.57 | 3.00±0.00 | 3.00±0.00 | 3.00±0.00 | 2.00±0.00 | 2.00±0.00 | 1.33±0.57 | **0.00±0.00** |
| | Survey | 2.00±0.00 | 4.00±0.00 | 5.00±0.00 | 5.00±0.00 | 3.00±0.00 | 3.33±0.69 | 2.00±0.00 | **1.00±0.00** |
| | Asia | 1.00±0.00 | 4.00±0.00 | 6.60±0.79 | 4.40±1.35 | 3.40±0.48 | 1.75±0.43 | 2.80±1.30 | **0.97±0.62** |
| | Asia-M | 2.00±0.00 | 4.00±0.00 | 7.60±0.48 | 4.60±0.48 | 3.20±0.39 | 1.68±0.46 | 2.00±0.00 | **0.00±0.00** |
| | Child | 8.25±0.00 | **3.00±0.00** | 12.6±0.79 | 10.8±1.72 | 14.2±0.40 | **3.00±0.00** | 7.00±0.00 | **3.00±0.00** |
| | Neuropathic | 8.62±0.00 | 6.00±0.00 | 9.33±0.94 | 10.0±0.00 | 10.0±0.00 | 4.20±0.96 | 9.00±0.00 | **1.23±0.42** |
| **N = 10000** | Earthquake | **0.00±0.00** | 4.00±0.00 | 3.00±0.00 | 3.00±0.00 | 1.00±0.00 | 0.40±0.48 | **0.00±0.00** | **0.00±0.00** |
| | Cancer | 2.00±0.00 | 3.00±0.00 | 3.00±0.00 | 3.00±0.00 | 2.00±0.00 | 0.60±0.80 | 2.00±0.00 | **0.00±0.00** |
| | Survey | 2.00±0.00 | 4.00±0.00 | 5.00±0.00 | 5.00±0.00 | 3.00±0.00 | 3.60±1.35 | 2.00±0.00 | **0.83±0.00** |
| | Asia | 1.50±0.00 | 4.00±0.00 | 6.00±0.00 | 4.40±1.35 | 3.00±0.00 | 1.40±0.48 | **0.00±0.00** | 0.34±0.47 |
| | Asia-M | 1.00±0.00 | 4.00±0.00 | 8.00±0.00 | 4.80±0.39 | 3.00±0.00 | 2.00±0.00 | **0.00±0.00** | **0.00±0.00** |
| | Child | 6.00±3.04 | 3.00±0.00 | 12.2±1.46 | 11.6±0.48 | 14.4±0.48 | 2.80±0.84 | 5.00±2.64 | **1.00±0.00** |
| | Neuropathic | 10.00±0.00 | 6.00±0.00 | **1.00±0.00** | 10.0±0.00 | 10.0±0.00 | 3.00±0.00 | 10.00±0.00 | **1.00±0.00** |

Table 1: Comparison with existing discovery methods. Mean and std dev of $D_{top}$ over 3 runs. (For Neuropathic subgraph (1k samples), PC Algorithm returns cyclic graphs in the MEC, therefore $D_{top}$ represented by '-' )

## 4.3 Order suitable for Expert Input than Edges

Causal order is more straightforward to elicit and evaluate objectively. Unlike edge existence, it is independent of other variables, ensuring greater consistency in expert responses. For example, in Fig. 3 (left), the causal link from *pollution* to *dyspnoea* via *cancer* highlights this distinction. The presence of an edge depends on whether *Cancer* is considered, leading to potential inconsistency in expert judgments. However, causal order remains consistent regardless of additional nodes, offering a more reliable assessment. Apriori, it is difficult to know which nodes may be relevant for a pair; hence experts' answers may not be consistent for questions about edges, but will be consistent on causal order.

## 5 LLMs as Virtual Experts for Causal Order

We now study whether LLMs can be used to obtain accurate causal order, thereby reducing the dependence on human experts for effect inference. We propose two prompting strategies based on utilizing variable names but not the associated data.

## 5.1 Pairwise Prompting Based Techniques

Recent work follows a pairwise strategy where each edge is inferred independently (Kıcıman et al. 2023; Ban et al. 2023; Long, Schuster, and Piché 2022). Our hypothesis is that adding context relevant to the pair of variables may help

increase the accuracy of the LLM answers. We propose four types of pairwise prompts (see Tables § A3 - § A10)

- **Basic prompt.** The simplest technique. We ask LLM to find the causal direction between a given pair of variables.

- **Iterative Context.** Here we provide the previously oriented pairs as context in the prompt. Since the LLM has access to its previous decisions, we expect that cycles will be limited through its predictions.

- **Markov Blanket Context.** Providing previously oriented pairs may become prohibitive for large graphs. Using the fact that a variable is independent of all other nodes given the Markov Blanket (Pearl 2009), we provide the Markov Blanket of the given node pairs as additional context.

- **Chain-of-Thought (+In-context learning).** Based on encouraging results of providing in-context examples in the prompt for various tasks (Brown et al. 2020), here we include examples of the causal orientation task that we expect the LLM to perform. Effectively, we provide example node pairs and their correct causal ordering before asking the question about the given nodes. Each example answer also contains its explanation, generated using Bing GPT-4. Adding explanations encourage LLM to employ chain-of-thought reasoning (Wei et al. 2022) when deciding the causal order.

## 5.2 Prompt Technique Based on Triplets

As we shall see, while pairwise prompts are conceptually simple, they are prone to yielding higher number of cycles in the graph since they decide about each edge separately. Taking inspiration from the PC algorithm that employs constraints over three variables, we now describe a prompting technique based on iterating over all possible triplets given a set of nodes. Once the LLM has provided subgraphs for each triplet, we determine causal order between a pair by aggregating over all triplet LLM answers where the pair was included.

The algorithm has the following steps after generating all possible triplets from a set of nodes. **1)** Generate subgraphs over all triplets through LLMs by prompting them to orient the three causal edges for each triplet. **2)** Merge the resultant structure between any two nodes by aggregating the number of LLM answers for the three orientations: (A → B; B → A; No connection) and choosing the majority answer. If there's a tie in edge orientation, GPT-4 is used with a CoT prompt to make the final decision. Then the causal order is extracted from the graph.

## 6 LLM-Guided Discovery Algorithms

Causal order from LLMs may exhibit unknown failure modes (Kıcıman et al. 2023). Hence we now provide algorithms for combining LLMs with causal discovery paradigms.

- **Constraint-based Algorithm**: Given a graph from constraint based algorithm like PC where some edges are not oriented, we use the causal order $\hat{\pi}$ from LLM to orient the undirected edges. Iterating over the undirected edges, we first check if the nodes of that edge are occurring in $\hat{\pi}$. If yes, we orient the edge according to the causal order. Since there is a possibility that LLM's final graph might have some isolated nodes which won't be in $\hat{\pi}$, therefore if either (or both) nodes of the undirected edge are not included in $\hat{\pi}$, we query GPT-4 using pairwise CoT prompt (from Sec. 5.1) to finalise a direction between the pair. Refer to Algorithm 1 in Appendix.
- **Score-based Algorithm**: We provide the level order of the causal graph returned by LLM as a prior for a score-based algorithm. Optionally, we can provide prior probability to control the influence of prior on the algorithm. Algorithm 2 in Appendix outlines the steps to combine score based method and the prior level order of variables.

## 7 Experiments and Results

To evaluate the accuracy of LLM-based algorithms on inferring causal order, we perform experiments on the benchmark datasets from Bayesian network repository (Scutari and Denis 2014): Earthquake, Cancer, Survey, Asia, Asia modified (Asia-M), and Child (see Appendix§ D for details). We also used a medium sized subset graph from the Neuropathic dataset (Tu et al. 2019) used for pain diagnosis.

**$D_{top}$ correlates with effect estimation error:** Before comparing methods on the $D_{top}$ metric, we first show that $D_{top}$ has a strong correlation with effect estimation error and hence is the correct metric for effect inference. Specifically, we

| Cancer | | | |
|---|---|---|---|
| $SHD$ vs. $\epsilon_{ACE} \mid D_{top} = 0$ | | $D_{top}$ vs. $\epsilon_{ACE} \mid SHD = 2$ | |
| $SHD$ | $\epsilon_{ACE}$ | $D_{top}$ | $\epsilon_{ACE}$ |
| 0 | 0.00 | 0 | 0.00 |
| 2 | 0.00 | 1 | 0.25 |
| 4 | 0.00 | 2 | 0.50 |
| **Asia** | | | |
| $SHD$ vs. $\epsilon_{ACE} \mid D_{top} = 0$ | | $D_{top}$ vs. $\epsilon_{ACE} \mid SHD = 3$ | |
| $SHD$ | $\epsilon_{ACE}$ | $D_{top}$ | $\epsilon_{ACE}$ |
| 0 | 0.00 | 1 | 0.14 |
| 6 | 0.00 | 2 | 0.22 |
| 10 | 0.00 | 3 | 0.57 |
| **Survey** | | | |
| $SHD$ vs. $\epsilon_{ACE} \mid D_{top} = 0$ | | $D_{top}$ vs. $\epsilon_{ACE} \mid SHD = 2$ | |
| $SHD$ | $\epsilon_{ACE}$ | $D_{top}$ | $\epsilon_{ACE}$ |
| 0 | 0.00 | 0 | 0.00 |
| 2 | 0.00 | 1 | 0.25 |
| 4 | 0.03 | 2 | 0.50 |

Table 2: $\epsilon_{ACE}$ vs. $SHD$ ($D_{top}$) given $D_{top}$ ($SHD$)

study how the error in causal effect, $\epsilon_{ACE}$, changes as values of the metrics $SHD, D_{top}$ change. In each graph, we evaluate causal effects of each variable on a specified target variable. We iterate through estimated causal graphs with different values of SHD and $D_{top}$ and report the mean absolute difference between estimated and true causal effects. As Table 2 shows, when $D_{top}$ is zero, effect error $\epsilon_{ACE}$ is also zero. And as $D_{top}$ increases (right panel), effect error increases. In contrast, SHD has no correlation with the $\epsilon_{ACE}$.

**Triplet prompting is most accurate for causal order:** Comparing prompting techniques (Tab. 3), we observe limitations with pairwise prompts as graph size increases. They often lead to significantly high cycles, making $D_{top}$ calculation infeasible. Notably, for the 20-node Child dataset, pairwise prompts result in thousands of cycles. Similar trends can be seen for Neuropathic graph as well. Among pairwise prompts, the chain-of-thought prompt achieves the lowest SHD for the small graphs and the fewest cycles for Child and Neuropathic. This highlights the effectiveness of in-context examples and chain-of-thought reasoning in improving causal order accuracy. The triplet prompt yields highly accurate causal order predictions. For all small graphs, number of cycles are 0 and $D_{top}$ is either 0 or 1. Additionally, the LLM output has lowest SHD. While the number of cycles increases when scaled to larger graphs (Child and Neuropathic), the numer is still significantly much smaller than cycles in pairwise output (all setups included).

Since pairwise orientations yielded substantially more cycles then triplet (see Table 3), we applied a cycle removal algorithm to triplet output only, to use it as prior for discovery algorithms. Our cycle removal algorithm is inspired from (Zheng et al. 2018). In the original approach, the algorithm minimizes edges to form a weighted DAG. As our noisy expert graphs lack edge weights, we leverage triplet pipeline votes to establish a probability distribution for edge orientations. Using this, we calculate the entropy for each

| Dataset | $D_{top}$ | SHD | IN/TN | Cycles |
|---|---|---|---|---|
| Base Prompt | | | | |
| Earthquake | 0 | 7 | 0/5 | 0 |
| Cancer | 0 | 6 | 0/5 | 0 |
| Survey | 3 | 12 | 0/6 | 0 |
| Asia | - | 21 | 0/8 | 1 |
| Asia-M | - | 15 | 0/7 | 7 |
| Child | - | 177 | 0/20 | >>3k |
| Neuropathic | - | 212 | 0/22 | >>5k |
| All Directed Edges | | | | |
| Earthquake | 1 | 9 | 0/5 | 0 |
| Cancer | 1 | 7 | 0/5 | 0 |
| Survey | 2 | 11 | 0/6 | 0 |
| Asia | - | 21 | 0/8 | 6 |
| Asia-M | 0 | 13 | 0/7 | 0 |
| Child | - | 139 | 0/20 | >>300 |
| Neuropathic | - | 194 | 0/22 | >>1k |
| Markov Blanket | | | | |
| Earthquake | 0 | 8 | 0/5 | 0 |
| Cancer | 0 | 6 | 0/5 | 0 |
| Survey | 3 | 12 | 0/6 | 0 |
| Asia | - | 21 | 0/8 | 1 |
| Asia-M | 0 | 14 | 0/7 | 0 |
| Child | - | 167 | 0/20 | >>400 |
| Neuropathic | - | 204 | 0/22 | >>4k |
| Chain of Thought | | | | |
| Earthquake | 0 | 4 | 0/5 | 0 |
| Survey | 1 | 9 | 2/6 | 0 |
| Asia | - | 18 | 0/8 | 1 |
| Asia-M | - | 13 | 0/7 | 1 |
| Child | - | 138 | 0/20 | >>500 |
| Neuropathic | - | 64 | 0/22 | >4 |
| Triplet Prompt | | | | |
| Earthquake | 0 | 4 | 0/5 | 0 |
| Cancer | 1 | 6 | 0/5 | 0 |
| Survey | 0 | 9 | 0/6 | 0 |
| Asia | 1 | 14 | 0/8 | 0 |
| Asia-M | 1 | 11 | 0/7 | 0 |
| Child | - | 138 | 0/20 | >63 |
| Child (+ Cycle Remover) | 1 | 28 | 10/20 | 0 |
| Neuropathic | - | 151 | 0/22 | >145 |
| Neuropathic(+ Cycle remover) | 3 | 24 | 13/20 | 0 |

Table 3: Comparison of various prompting strategies for only LLM based setups. IN: Isolated Nodes, TN:Total Nodes. CoT (Cancer not included since CoT prompt has examples from this graph). Calculating total cycles in a DAG is NP-Hard. We estimate a lower bound using cycles of length k (k=5). Scaling k increases unique cycles in the graph significantly. However, Triplet setup gives significantly lesser cycles, with total number of cycles for Child = 391 and Neuropathic = 772.

edge, removing those with higher entropy (lower confidence). To minimize $D_{top}$, we pruned edges with entropy below the mean of all entropies. However, optimizing the threshold and minimizing edges for a DAG increased connected nodes but also led to a higher $D_{top}$, diminishing the quality of the prior and causing poor causal discovery performance. Since pruning edges to remove cycles in the pairwise case would have resulted in a non-significant prior, we only apply this on triplet prompting where cycle removal is feasible.

**LLMs improve causal order accuracy of existing discovery algorithms:** We investigate if LLM output enhances causal order inference accuracy in discovery algorithms. We compare against widely used methods: PC (Spirtes, Glymour, and Scheines 2000), SCORE (Rolland et al. 2022), ICA-LiNGAM (Shimizu et al. 2006), Direct-LiNGAM (Shimizu et al. 2011), NOTEARS (Zheng et al. 2018), and CaMML (Wallace, Korb, and Dai 1996) across sample sizes: $250, 500, 1000, 5000, 10000$ (refer Table 1 for full results). We employ the triplet prompt with LLM. Table 1 presents the $D_{top}$ metric for different algorithms and compares it with PC+LLM and CaMML+LLM. PC and CaMML exhibit superior performance with the lowest $D_{top}$ among the discovery algorithms.LLM output reduces $D_{top}$ in both algorithms. PC+LLM exhibits substantial improvements, particularly at lower sample sizes, implying LLM's significance in limited data settings. Transitioning from CaMML to CaMML+LLM also shows significant $D_{top}$ reductions, with benefits even at higher sample sizes. For instance, at sample size 10000, CaMML+LLM outperforms CaMML by a factor of three for Child and five for Asia. These findings underscore the substantial enhancement LLMs provide to causal discovery algorithms.

# 8 Limitations and Conclusions

We presented causal order as a suitable metric for evaluating quality of causal graphs for downstream effect inference tasks. Using a novel formulation of LLM prompts based on triplets, we showed that LLMs can be useful in the generating accurate causal order, both individually and in combination with existing discovery algorithms. That said, one limitation is that we studied LLMs utility on popular benchmarks which may have been partially memorized. It will be interesting to extend our experiments to more datasets and tasks. Also, we focused on one causal task, viz. effect inference, in this work; identifying suitable metrics for tasks such as causal prediction and counterfactual inference and extending to other tasks in causal inference may be useful directions for future work.

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

# Appendix

## A Proofs of Propositions

**Proposition A.1.** *(Pearl 2009; Cinelli, Forney, and Pearl 2022a) Under the no-latent confounding assumption, given a pair of treatment and target variables $(X_i, X_j)$, $\mathbf{Z} = \{X_k | \pi_k < \pi_i\}$ is a valid adjustment set relative to $(X_i, X_j)$ for any topological order $\pi$ of nodes $X_i, \ldots, X_n$.*

*Proof.* We need to show that the set $\mathbf{Z} = \{X_k | \pi_k < \pi_i\}$ satisfies the conditions (i) and (ii) in the definition of backdoor adjustment set. For any variable $X_k$ such that $\pi_k < \pi_i$, we have $X_k \notin de(X_i)$ and hence the condition (i) is satisfied. Additionally, for each $X_k \in pa(X_i)$ we have $\pi_k < \pi_i$ and hence $pa(X_i) \subseteq \mathbf{Z}$. Since $pa(X_i)$ blocks all paths from $X_i$ to $X_j$ that contains an arrow into $X_i$ (Peters and Bühlmann 2015), $\mathbf{Z}$ satisfies condition (ii). $\square$

**Proposition A.2.** *For an estimated topological order $\hat{\pi}$ and a true topological order $\pi$ of a causal DAG $\mathcal{G}$ with the corresponding adjacency matrix $A$, $D_{top}(\hat{\pi}, A) = 0$ iff $\mathbf{Z} = \{X_k | \hat{\pi}_k < \hat{\pi}_i\}$ is a valid adjustment set relative to $(X_i, X_j)$, $\forall \pi_i < \pi_j$.*

*Proof.* The statement of proposition is of the form $A \iff B$ with $A$ being "$D_{top}(\hat{\pi}, A) = 0$" and $B$ being "$\mathbf{Z} = \{X_k | \hat{\pi}_k < \hat{\pi}_i\}$ is a valid adjustment set relative to $(X_i, X_j)$, $\forall i, j$". We prove $A \iff B$ by proving (i) $A \implies B$ and (ii) $B \implies A$.

(i) Proof of $A \implies B$: If $D_{top}(\hat{\pi}, A) = 0$, for all pairs of nodes $(X_i, X_j)$, we have $\hat{\pi}_i < \hat{\pi}_j$ whenever $\pi_i < \pi_j$. That is, causal order in estimated graph is same that of the causal order in true graph. Hence, from Propn 4.1, $\mathbf{Z} = \{X_k | \hat{\pi}_k < \hat{\pi}_i\}$ is a valid adjustment set relative to $(X_i, X_j)$, $\forall i, j$.

(ii) Proof of $B \implies A$: we prove the logical equivalent form of $B \implies A$ i.e., $\neg A \implies \neg B$, the *contrapositive* of $B \implies A$. To this end, assume $D_{top}(\hat{\pi}, A) \neq 0$, then there will be at least one edge $X_i \rightarrow X_j$ that cannot be oriented correctly due to the estimated topological order $\hat{\pi}$. i.e., $\hat{\pi}_j < \hat{\pi}_i$ but $\pi_j > \pi_i$. Hence, to find the causal effect of $X_i$ on $X_l$; $l \neq j$, $X_j$ is included in the back-door adjustment set $\mathbf{Z}$ relative to $(X_i, X_l)$. Adding $X_j$ to $\mathbf{Z}$ renders $\mathbf{Z}$ an invalid adjustment set because it violates the condition (i). $\square$

**Proposition A.3.** *In a causal DAG $\mathcal{G}$ with $N$ levels in the level-ordering of variables where the level $i$ contains $n_i$ variables, $\exists \hat{\mathcal{G}}$ s.t. $SHD(\hat{\mathcal{G}}, \mathcal{G}) \geq \sum_{i=1}^{N-1} (n_i \times \sum_{j=i+1}^{N} n_j) - |\mathbf{E}|$ and $D_{top}(\hat{\pi}, A) = 0 \; \forall \hat{\pi}$ of $\hat{\mathcal{G}}$.*

*Proof.* Recall that SHD counts the number of missing, falsely detected, and falsely directed edges in the estimated causal graph as compared to the ground truth graph. Since we want $D_{top}(\hat{\pi}, A) = 0$; $\forall \hat{\pi}$ of $\hat{\mathcal{G}}$, there cannot be an edge $X_i \rightarrow X_j$ in $\hat{\mathcal{G}}$ such that $X_i \leftarrow X_j$ is in $\mathcal{G}$. This constraint avoids the possibility of having falsely directed edges in $\hat{\mathcal{G}}$. Consider a $\hat{\mathcal{G}}$ with all the edges in $\mathcal{G}$ and in addition, each variable in level $i$ having a directed edge to each variable in all levels below level $i$. All such edges contribute to the SHD score

while still obeying the causal ordering in $\mathcal{G}$. This number will be equal to $\sum_{i=1}^{N-1} (n_i \times \sum_{j=i+1}^{N} n_j) - |\mathbf{E}|$. The quantity $\sum_{i=1}^{N-1} (n_i \times \sum_{j=i+1}^{N} n_j)$ is the number of edges possible from each node to the every other node in the levels below it. We need to subtract the number of existing edges in $\mathbf{E}$ to count the newly added edges that contribute to the SHD score. Now, we can remove some of the edges $X_i \rightarrow X_j$ from $\hat{\mathcal{G}}$ such that $X_i \rightarrow X_j$ is in $\mathcal{G}$ while still leading to same causal ordering of variables $\hat{\mathcal{G}}$. This leads to increased SHD score due to missing edges in $\hat{\mathcal{G}}$. Since it will only increase the SHD score, we ignore such corner cases. $\square$

## B Additional Results

Tab. 2 shows the correlation between $D_{top}$ and $\epsilon_{ACE}$, where for the datasets Cancer, Asia and Survey, we consider $dyspnoea$, $dyspnoea$, and $Travel$ respectively as the target variables. Whereas, there is no correlation between SHD and $\epsilon_{ACE}$.

Each experiment was conducted three times, with reported results as mean and standard deviation values. In experiments involving Child and Insurance datasets (Algorithm 2), we averaged over prior probabilities $0.5, 0.6, 0.7, 0.8, 0.9$. Table 1 demonstrates the superior performance of our proposed methods compared to the baselines.

### B.1 LLMs used in post-processing for graph discovery

We conducted some experiments where we utilised discovery algorithms like PC for creating skeletons of the graph and employed LLMs for orienting the undirected edges. The idea was to utilise LLMs ability to correctly estimate the causal direction while leveraging PC algorithm's ability to give a skeleton which could be oriented in a post processing setup. We saw that LLM ended up giving improved results as compared to PC alone.

| Context ($\rightarrow$) | Base prompt | Undirected graph | Past iteration orientations | Markov Blanket | PC (Average over MEC) |
|---|---|---|---|---|---|
| | | | 1000 samples | | |
| $D_{top}$ | 8.0 | 7.0 | 5.3 | 6.6 | 9.61 |
| SHD | 14.33 | 14.33 | 12.66 | 14.0 | 17.0 |
| | | | 10000 samples | | |
| $D_{top}$ | 6.33 | 8.33 | 9.66 | 6.0 | 7.67 |
| SHD | 9.0 | 11.0 | 13.33 | 8.33 | 12.0 |

Table A1: PC + LLM results where LLM is used to orient the undirected edges of the skeleton PC returns over different data sample sizes. We show how LLMs can be used in a post processing setup for edge orientation besides having the capability of acting as a strong prior for different discovery algorithm

## C Algorithms

Algorithms 1 and 2 outlines the procedures to combine LLMs with existing constraint-based and score-based methods.

**Algorithm 1:** Combining constraint based methods and experts to get $\hat{\pi}$ for a given set of variables.

---

1: **Input:** LLM topological ordering $\hat{\pi}$, Expert $\mathcal{E}_{GPT4}$, PC-CPDAG $\hat{\mathcal{G}}$
2: **Output:** Estimated topological order $\hat{\pi}_{\text{final}}$ of $\{X_1, \ldots, X_n\}$.
3: **for** $(i - j) \in$ undirected-edges($\hat{\mathcal{G}}$) **do**
4:      If both the node $i$ and $j$ are in $\hat{\pi}$ and if $\hat{\pi}_i < \hat{\pi}_j$, orient $(i-j)$ as $(i \rightarrow j)$ in $\hat{\mathcal{G}}$.
5:      Otherwise, use the expert $\mathcal{E}_{GPT4}$ with CoT prompt to orient the edge $(i-j)$.
6: **end for**
7: $\hat{\pi}_{\text{final}} =$ topological ordering of $\hat{\mathcal{G}}$
8: **return** $\hat{\pi}$

---

**Algorithm 2:** Combining score based methods and experts to get $\hat{\pi}$ for a given set of variables.

---

1: **Input:** $\mathcal{D}$, variables $\{X_1, \ldots, X_n\}$, Expert $\mathcal{E}$, Score based method $\mathcal{S}$, *Prior* probability $p$.
2: **Output:** Estimated topological order $\hat{\pi}$ of $\{X_1, \ldots, X_n\}$.
3: Step (I) $\hat{\mathcal{G}} = \mathcal{E}(X_1, \ldots, X_n)$
4: Step (II) *Prior* = level order traversal of $\hat{\mathcal{G}}$.
5: Step (II.I) If $\hat{\mathcal{G}}$ is cyclic, keep all the variables in a cycle at the same level in *Prior*.
6: Step (III) $\hat{\mathcal{G}} = \mathcal{S}(\mathcal{D}, Prior, Prior\ probability = p)$
7: Step (IV) $\hat{\pi} =$ topological ordering of $\hat{\mathcal{G}}$
8: **return** $\hat{\pi}$

---

## D  Causal Graphs used in Experiments

Figs. A1-A5 show the causal graphs and details we considered from BNLearn repository (Scutari and Denis 2014) (Tab. A2). To construct Asia-M From the Asia graph, we first remove the node *either* and its corresponding edges. We then add directed edges from the parents of *either* to the children of *either* to preserve the original causal order in Asia-M. Tables A3-A12 show the prompt structures and examples we study.

| Dataset | # of Nodes | # of Edges | Description (used as a context) |
|---|---|---|---|
| Asia | 8 | 8 | Model the possible respiratory problems someone can have who has recently visited Asia and is experiencing shortness of breath |
| Cancer | 5 | 4 | Model the relation between various variables responsible for causing Cancer and its possible outcomes |
| Earthquake | 5 | 5 | Model factors influencing the probability of a burglary |
| Survey | 6 | 6 | Model a hypothetical survey whose aim is to investigate the usage patterns of different means of transport |
| Child | 20 | 25 | Model congenital heart disease in babies |

Table A2: Overview of the datasets used.

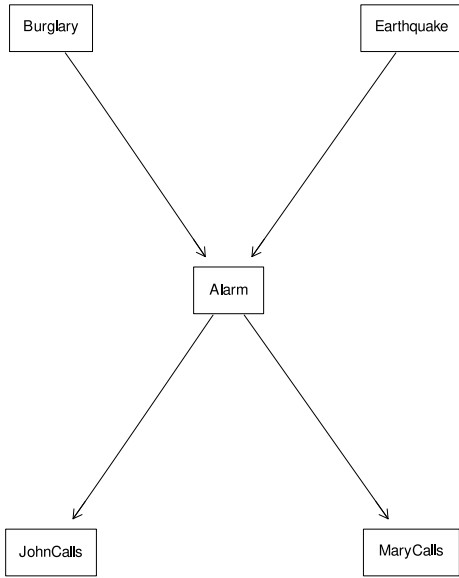

Figure A1: Earthquake Bayesian network. Abbreviations/Descriptions: Burglary: *burglar entering*, Earthquake: *earthquake hitting*, Alarm: *home alarm going off in a house*, JohnCalls: *first neighbor to call to inform the alarm sound*, Marycalls: *second neighbor to call to inform the alarm sound*.

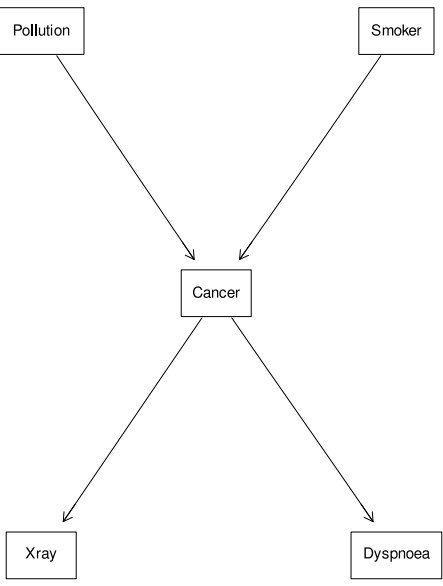

Figure A2: Cancer Bayesian network. Abbreviations/Descriptions: Pollution: *exposure to pollutants*, Smoker: *smoking habit*, Cancer: *Cancer*, Dyspnoea: *Dyspnoea*, Xray: *getting positive xray result*.

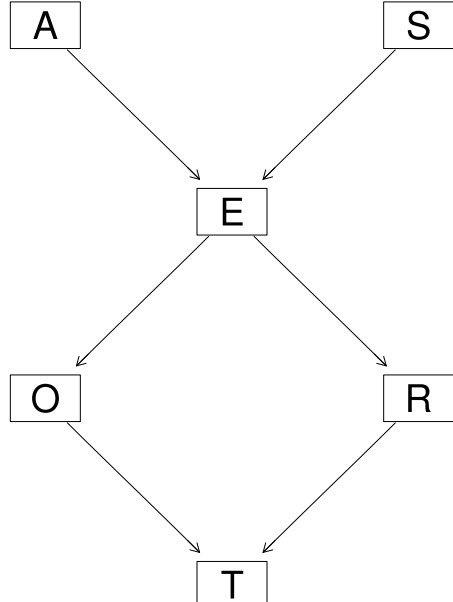

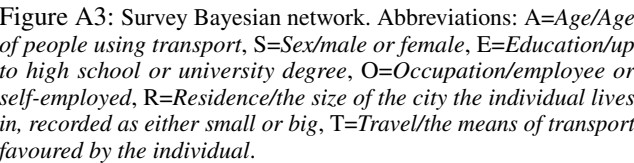

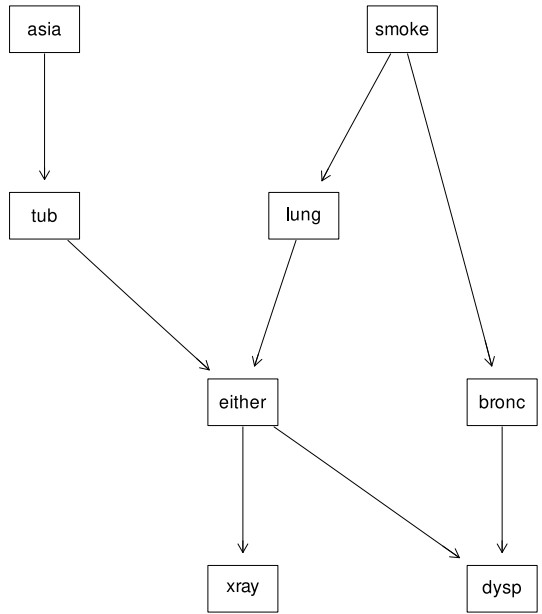

Figure A3: Survey Bayesian network. Abbreviations: A=*Age/Age of people using transport*, S=*Sex/male or female*, E=*Education/up to high school or university degree*, O=*Occupation/employee or self-employed*, R=*Residence/the size of the city the individual lives in, recorded as either small or big*, T=*Travel/the means of transport favoured by the individual*.

Figure A4: Asia Bayesian network. Abbreviations/Descriptions: asia=*visit to Asia/visiting Asian countries with high exposure to pollutants*, smoke=*smoking habit*, tub=*tuberculosis*, lung=*lung cancer*, either=*either tuberculosis or lung cancer*, bronc=*bronchitis*, dysp=*dyspnoea*, xray=*getting positve xray result*.

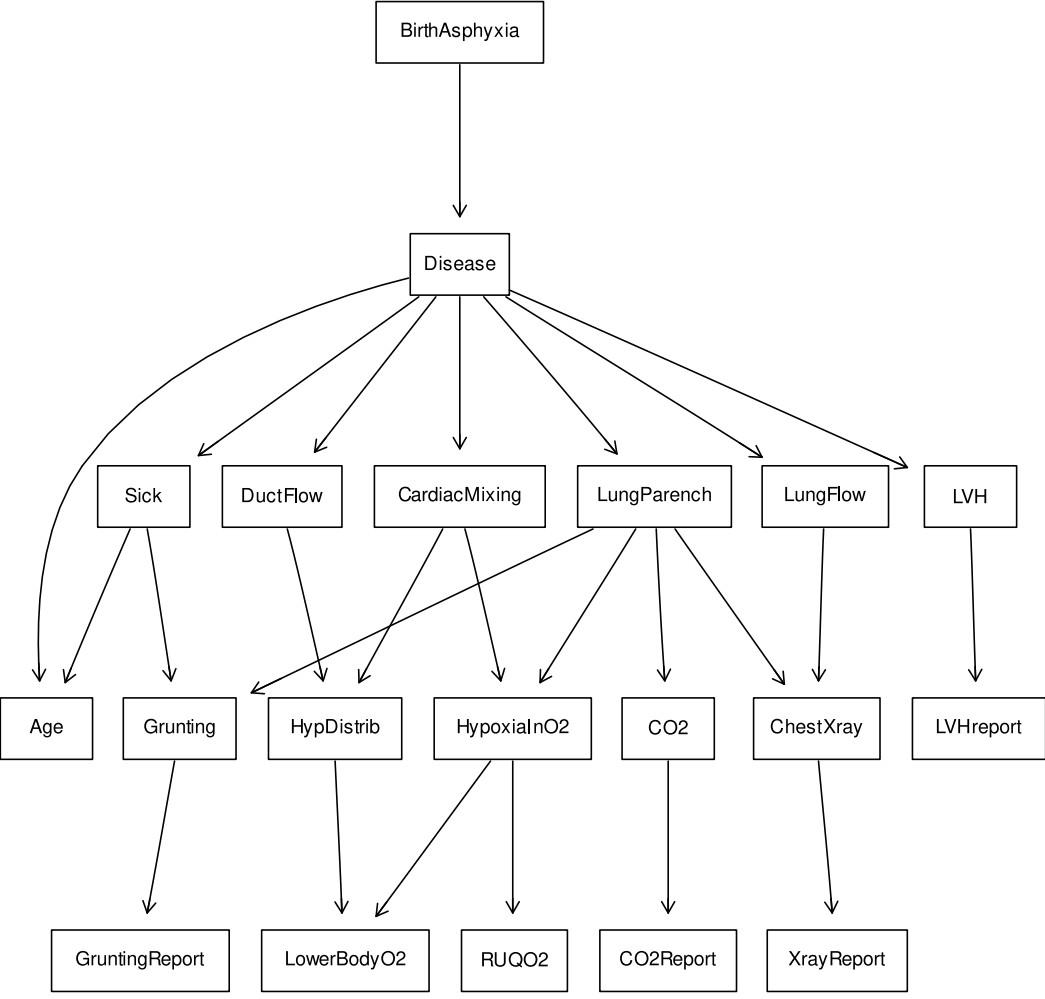

Figure A5: Child Bayesian network. Abbreviations: BirthAsphyxia: *Lack of oxygen to the blood during the infant's birth*, HypDistrib: *Low oxygen areas equally distributed around the body*, HypoxiaInO2: *Hypoxia when breathing oxygen*, CO2: *Level of carbon dioxide in the body*, ChestXray: *Having a chest x-ray*, Grunting: *Grunting in infants*, LVHreport: *Report of having left ventricular hypertrophy*, LowerBodyO2: *Level of oxygen in the lower body*, RUQO2: *Level of oxygen in the right upper quadricep muscle*, CO2Report: *A document reporting high levels of CO2 levels in blood*, XrayReport: *Report of having a chest x-ray*, Disease: *Presence of an illness*, GruntingReport: *Report of infant grunting*, Age: *Age of infant at disease presentation*, LVH: *Thickening of the left ventricle*, DuctFlow: *Blood flow across the ductus arteriosus*, CardiacMixing: *Mixing of oxygenated and deoxygenated blood*, LungParench: *The state of the blood vessels in the lungs*, LungFlow: *Low blood flow in the lungs*, Sick: *Presence of an illness*

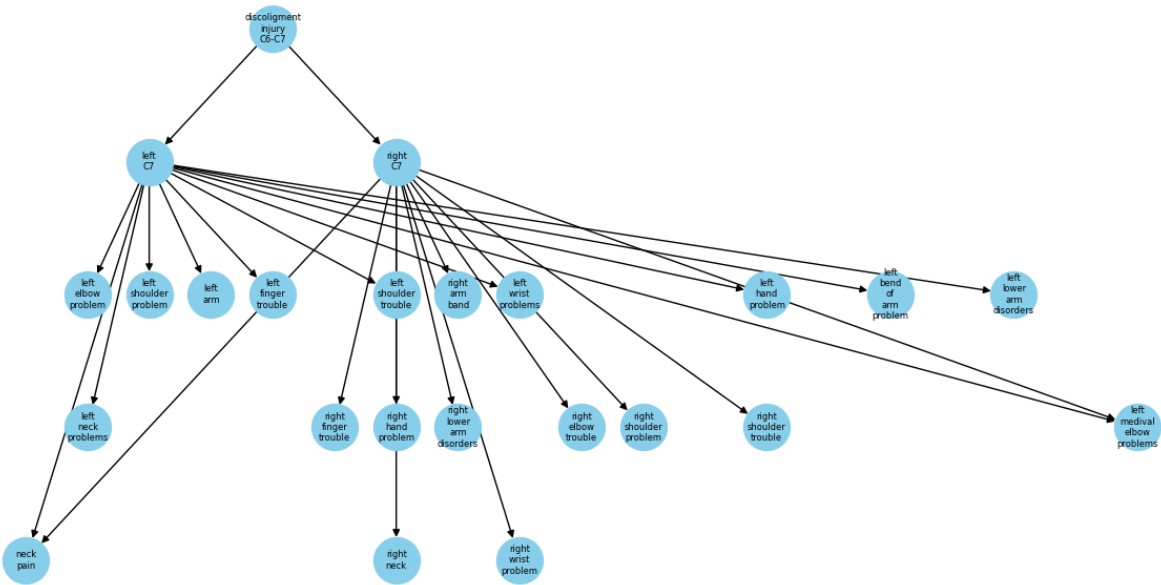

Figure A6: For Neuropathic dataset, we consider a sub-graph induced by one of the root nodes, containing the following 22 nodes and corresponding edges taken from https://observablehq.com/@turuibo/the-complete-causal-graph-of-neuropathic-pain-diagnosis: 'right C7', 'right elbow trouble', 'left shoulder trouble', 'left bend of arm problem', 'right shoulder trouble', 'right hand problem', 'left medival elbow problems', 'right finger trouble', 'left neck problems', 'left wrist problems', 'left shoulder problem', 'right neck', 'right wrist problem', 'right shoulder problem', 'discoligment injury C6 C7', 'left hand problem', 'left C7', 'right arm band', 'left lower arm disorders', 'neck pain', 'left finger trouble', 'left arm'. We did not use descriptions for the nodes of Neuropathic graph.

Question: For a causal graph used to model relationship of various factors and outcomes related to cancer with the following nodes: ['Pollution', 'Cancer', 'Smoker', 'Xray', 'Dyspnoea'],
Which cause-and-effect relationship is more likely between nodes 'smoker' and 'cancer'?

A. changing the state of node 'smoker' causally effects a change in another node 'cancer'.

B. changing the state of node 'cancer' causally effects a change in another node 'smoker'.

C. There is no causal relation between the nodes 'cancer' and 'smoker'.

Make sure to first provide a grounded reasoning for your answer and then provide the answer in the following format: <Answer>A/B/C</Answer>. It is very important that you output the final Causal graph within the tags like <Answer>A/B/C</Answer> otherwise your answer will not be processed.

Answer: The causal effect of "smoker" directing to "cancer" is based on the strong evidence from epidemiological studies linking smoking to an increased risk of developing cancer. Smoking introduces harmful substances into the respiratory system, leading to cellular damage and mutation, which significantly raises the likelihood of cancer development in the lungs or respiratory tract, subsequently impacting the occurrence of respiratory problems like shortness of breath. Therefore answer is <Answer>A</Answer>

Question: For a causal graph used to model relationship of various factors and outcomes related to cancer with the following nodes: ['Pollution', 'Cancer', 'Smoker', 'Xray', 'Dyspnoea'], Which cause-and-effect relationship is more likely between nodes 'xray' and 'dyspnoea'?

A. changing the state of node 'xray' causally effects a change in another node 'dyspnoea'.

B. changing the state of node 'dyspnoea' causally effects a change in another node 'xray'.

C. There is no causal relation between the nodes 'xray' and 'dyspnoea'.

Make sure to first provide a grounded reasoning for your answer and then provide the answer in the following format: <Answer>A/B/C</Answer>. It is very important that you output the final Causal graph within the tags like <Answer>A/B/C</Answer> otherwise your answer will not be processed.

Answer: Reasoning behind the lack of causal relation between X-ray and dyspnoea is that X-ray and dyspnoea are both effects of having cancer, but they do not directly cause or affect each other. X-ray is a diagnostic test that can help detect cancer in the lungs or other organs, while dyspnoea is a symptom of cancer that involves feeling short of breath. Therefore, X-ray and dyspnoea are not causally related, but they are both associated with cancer. Therefore answer is <Answer>C</Answer>

Question: For a causal graph used to model relationship of various factors and outcomes related to cancer with the following nodes: ['Pollution', 'Cancer', 'Smoker', 'Xray', 'Dyspnoea'], Which cause-and-effect relationship is more likely between nodes 'xray' and 'cancer'?

A. changing the state of node 'xray' causally effects a change in another node 'cancer'.

B. changing the state of node 'cancer' causally effects a change in another node 'xray'.

C. There is no causal relation between the nodes 'xray' and 'cancer'.

Make sure to first provide a grounded reasoning for your answer and then provide the answer in the following format: <Answer>A/B/C</Answer>. It is very important that you output the final Causal graph within the tags like <Answer>A/B/C</Answer> otherwise your answer will not be processed.

Table A3: Chain of Thought Prompt

Answer: The causal effect of cancer on X-ray is that X-rays are often used to diagnose or detect cancer in different parts of the body, such as the bones, lungs, breasts, or kidneys123. X-rays use low doses of radiation to create pictures of the inside of the body and show the presence, size, and location of tumors. X-rays can also help monitor the response to treatment or check for signs of recurrence. Therefore, having cancer may increase the likelihood of getting an X-ray as part of the diagnostic process or follow-up care. Therefore answer is <Answer>B</Answer>

Question: For a causal graph used to model relationship of various factors and outcomes related to cancer with the following nodes: ['Pollution', 'Cancer', 'Smoker', 'Xray', 'Dyspnoea'], Which cause-and-effect relationship is more likely between nodes 'pollution' and 'cancer'?

A. changing the state of node 'pollution' causally effects a change in another node 'cancer'.

B. changing the state of node 'cancer' causally effects a change in another node 'pollution'.

C. There is no causal relation between the nodes 'pollution' and 'cancer'.

Make sure to first provide a grounded reasoning for your answer and then provide the answer in the following format: <Answer>A/B/C</Answer>. It is very important that you output the final Causal graph within the tags like <Answer>A/B/C</Answer> otherwise your answer will not be processed.

Answer: The causal effect of pollution on cancer is that air pollution contains carcinogens (cancer-causing substances) that may be absorbed into the body when inhaled and damage the DNA of cells. Another possible reasoning is that particulate matter (tiny dust-like particles) in air pollution may cause physical damage to the cells in the lungs, leading to inflammation and oxidative stress and eventually cell mutations. A third possible reasoning is that air pollution may create an inflamed environment in the lungs that encourages the proliferation of cells with existing cancer-driving mutations. These are some of the hypotheses that researchers have proposed to explain how air pollution may cause cancer, but more studies are needed to confirm them. Therefore answer is <Answer>A</Answer>

Question: For a causal graph used to model relationship of various factors and outcomes related to cancer with the following nodes: ['Pollution', 'Cancer', 'Smoker', 'Xray', 'Dyspnoea'], Which cause-and-effect relationship is more likely between nodes 'pollution' and 'smoker'?

A. changing the state of node 'pollution' causally effects a change in another node 'smoker'.

B. changing the state of node 'smoker' causally effects a change in another node 'pollution'.

C. There is no causal relation between the nodes 'pollution' and 'smoker'.

Make sure to first provide a grounded reasoning for your answer and then provide the answer in the following format: <Answer>A/B/C</Answer>. It is very important that you output the final Causal graph within the tags like <Answer>A/B/C</Answer> otherwise your answer will not be processed.

Answer: Reason behind the lack of causal relation between pollution and smoker is that pollution and smoking are both independent risk factors for respiratory problems, but they do not directly cause or affect each other. Pollution and smoking both contribute to air pollution, which can harm the health of people and the environment. However, pollution is mainly caused by human activities such as burning fossil fuels, deforestation, or industrial processes, while smoking is a personal choice that involves inhaling tobacco smoke. Therefore, pollution and smoker are not causally related, but they are both associated with respiratory problems. Therefore answer is <Answer>C</Answer>.

Table A4: Chain of Thought Prompt (continued..)

Question: For a causal graph used for modeling factors causing Coronary Heart Diseases with the following nodes: ['Family Disease', 'Gene', 'Smoking', 'Blood Pressure', 'Coronary Heart Disease', 'Headache'], Which cause-and-effect relationship is more likely between nodes 'Family Disease' and 'Gene'?

A. changing the state of node 'Family Disease' causally effects a change in another node 'Gene'.

B. changing the state of node 'Gene' causally effects a change in another node 'Family Disease'.

C. There is no causal relation between the nodes 'Family Disease' and 'Gene'.

Make sure to first provide a grounded reasoning for your answer and then provide the answer in the following format: <Answer>A/B/C</Answer>. It is very important that you output the final Causal graph within the tags like <Answer>A/B/C</Answer> otherwise your answer will not be processed.

Answer: Reason behind the causal effect of family disease on gene is that family disease is a term that refers to diseases or health conditions that run in the family, meaning that they are influenced by genetic factors. Gene is a term that refers to the basic unit of heredity that carries information for a specific trait or function. Family disease can affect gene by altering the type or frequency of genes that are inherited by the offspring from their parents. For example, some family diseases are caused by deterministic genes, which are genes that guarantee the development of a disease if they are present in a person's genome. Other family diseases are influenced by risk genes, which are genes that increase the likelihood of developing a disease but do not guarantee it. Therefore, family disease can causally direct to gene by changing the genetic makeup of a person or a population. Therefore answer is <Answer>A</Answer>.

Question: For a causal graph used for modeling factors causing Coronary Heart Diseases with the following nodes: ['Family Disease', 'Gene', 'Smoking', 'Blood Pressure', 'Coronary Heart Disease', 'Headache'], Which cause-and-effect relationship is more likely between nodes 'Coronary Heart Disease' and 'Gene'?

A. changing the state of node 'Coronary Heart Disease' causally effects a change in another node 'Gene'.

B. changing the state of node 'Gene' causally effects a change in another node 'Coronary Heart Disease'.

C. There is no causal relation between the nodes 'Coronary Heart Disease' and 'Gene'.

Make sure to first provide a grounded reasoning for your answer and then provide the answer in the following format: <Answer>A/B/C</Answer>. It is very important that you output the final Causal graph within the tags like <Answer>A/B/C</Answer> otherwise your answer will not be processed.

Answer: Possible reasoning behind the causal effect of gene on coronary heart disease is that gene is a term that refers to the basic unit of heredity that carries information for a specific trait or function. Gene can affect coronary heart disease by influencing the structure and function of the blood vessels, the metabolism and transport of lipids (fats) in the blood, the inflammation and clotting processes, or the response to environmental factors such as smoking or diet. For example, some genes code for proteins that regulate the cell cycle and growth of the cells that line the arteries, which can affect their susceptibility to damage or plaque formation. Other genes code for proteins that control the synthesis and clearance of cholesterol or other lipids, which can affect their levels and deposition in the arteries. Therefore, gene can causally direct to coronary heart disease by modifying the biological pathways that contribute to the development or progression of the disease. Therefore answer is <Answer>B</Answer>

Table A5: Chain of Thought Prompt (continued..)

Question: For a causal graph used for modeling factors causing Coronary Heart Diseases with the following nodes: ['Family Disease', 'Gene', 'Smoking', 'Blood Pressure', 'Coronary Heart Disease', 'Headache'], Which cause-and-effect relationship is more likely between nodes 'Blood Pressure' and 'Smoking'?

A. changing the state of node 'Blood Pressure' causally effects a change in another node 'Smoking'.

B. changing the state of node 'Smoking' causally effects a change in another node 'Blood Pressure'.

C. There is no causal relation between the nodes 'Blood Pressure' and 'Smoking'.

Make sure to first provide a grounded reasoning for your answer and then provide the answer in the following format: <Answer>A/B/C</Answer>. It is very important that you output the final Causal graph within the tags like <Answer>A/B/C</Answer> otherwise your answer will not be processed.

Answer: Possible reasoning behind the causal effect of smoking on blood pressure is that smoking is a habit that involves inhaling tobacco smoke, which contains nicotine and other harmful chemicals. Smoking can affect blood pressure by activating the sympathetic nervous system (SNS), which is the part of the nervous system that controls the body's response to stress or danger. When the SNS is activated, it releases hormones such as adrenaline and noradrenaline, which cause the heart to beat faster and harder, and the blood vessels to constrict. This results in a temporary increase in blood pressure, which can last for 15 to 20 minutes after each cigarette. Therefore, smoking can causally direct to blood pressure by stimulating the SNS and increasing the cardiac output and vascular resistance. Therefore answer is <Answer>B</Answer>.

Question: For a causal graph used for modeling factors causing Coronary Heart Diseases with the following nodes: ['Family Disease', 'Gene', 'Smoking', 'Blood Pressure', 'Coronary Heart Disease', 'Headache'], Which cause-and-effect relationship is more likely between nodes 'Headache' and 'Smoking'?

A. changing the state of node 'Headache' causally effects a change in another node 'Smoking'.

B. changing the state of node 'Smoking' causally effects a change in another node 'Headache'.

C. There is no causal relation between the nodes 'Headache' and 'Smoking'.

Make sure to first provide a grounded reasoning for your answer and then provide the answer in the following format: <Answer>A/B/C</Answer>. It is very important that you output the final Causal graph within the tags like <Answer>A/B/C</Answer> otherwise your answer will not be processed.

Answer: One possible reasoning behind the lack of causal relation between headache and smoking is that headache and smoking are both associated with various health conditions, but they do not directly cause or affect each other12. Headache is a term that refers to pain or discomfort in the head, scalp, or neck, which can have many possible causes, such as stress, dehydration, infection, injury, or medication. Smoking is a habit that involves inhaling tobacco smoke, which contains nicotine and other harmful chemicals, which can increase the risk of diseases such as cancer, heart disease, stroke, and lung disease. Therefore, headache and smoking are not causally related, but they are both linked to different health problems. Therefore the answer is <Answer>C</Answer>

Table A6: Chain of Thought Prompt (continued..)

Question: For a causal graph used for modeling factors causing Coronary Heart Diseases with the following nodes: ['Family Disease', 'Gene', 'Smoking', 'Blood Pressure', 'Coronary Heart Disease', 'Headache'], Which cause-and-effect relationship is more likely between nodes 'Headache' and 'Smoking'?

A. changing the state of node 'Headache' causally effects a change in another node 'Smoking'.

B. changing the state of node 'Smoking' causally effects a change in another node 'Headache'.

C. There is no causal relation between the nodes 'Headache' and 'Smoking'.

Make sure to first provide a grounded reasoning for your answer and then provide the answer in the following format: <Answer>A/B/C</Answer>. It is very important that you output the final Causal graph within the tags like <Answer>A/B/C</Answer> otherwise your answer will not be processed.

Answer: One possible reasoning behind the lack of causal relation between headache and smoking is that headache and smoking are both associated with various health conditions, but they do not directly cause or affect each other. Headache is a term that refers to pain or discomfort in the head, scalp, or neck, which can have many possible causes, such as stress, dehydration, infection, injury, or medication. Smoking is a habit that involves inhaling tobacco smoke, which contains nicotine and other harmful chemicals, which can increase the risk of diseases such as cancer, heart disease, stroke, and lung disease. Therefore, headache and smoking are not causally related, but they are both linked to different health problems. Therefore the answer is <Answer>C</Answer>

Question: For a causal graph used for modeling factors causing Coronary Heart Diseases with the following nodes: ['Family Disease', 'Gene', 'Smoking', 'Blood Pressure', 'Coronary Heart Disease', 'Headache'], Which cause-and-effect relationship is more likely between nodes 'Coronary Heart Disease' and 'Smoking'?

A. changing the state of node 'Smoking' causally effects a change in another node 'Coronary Heart Disease'.

B. changing the state of node 'Coronary Heart Disease' causally effects a change in another node 'Smoking'.

C. There is no causal relation between the nodes 'Coronary Heart Disease' and 'Smoking'.

Make sure to first provide a grounded reasoning for your answer and then provide the answer in the following format: <Answer>A/B/C</Answer>. It is very important that you output the final Causal graph within the tags like <Answer>A/B/C</Answer> otherwise your answer will not be processed.

Answer: Possible reasoning behind the causal effect of smoking on coronary heart disease is smoking damages the heart and blood vessels by raising triglycerides, lowering HDL, increasing blood clotting, and impairing blood flow to the heart. This can lead to plaque buildup, heart attacks, and death. Therefore answer is <Answer>A</Answer>.

Question: For a causal graph used for context with the following nodes: nodes, Which cause-and-effect relationship is more likely between nodes X and Y?

A. changing the state of node X causally effects a change in another node Y.

B. changing the state of node Y causally effects a change in another node X.

C. There is no causal relation between the nodes X and Y.

Make sure to first provide a grounded reasoning for your answer and then provide the answer in the following format: <Answer>A/B/C</Answer>. It is very important that you output the final Causal graph within the tags like <Answer>A/B/C</Answer> otherwise your answer will not be processed.

Table A7: Chain of Thought Prompt (continued..)

Which cause-and-effect relationship is more likely?

A. changing the state of node which says X causally effects a change in another node which says Y.

B. changing the state of node which says Y causally effects a change in another node which says X.

C. There is no causal relationship between node X and Y.

Make sure to first output a factually grounded reasoning for your answer. X and Y are nodes of a Causal Graph. The causal graph is sparse and acyclic in nature. So option C could be chosen if there is some uncertainity about causal relationship between X and Y.

First give your reasoning and after that please make sure to provide your final answer within the tags <Answer>A/B/C</Answer>.
It is very important that you output your final answer between the tags like <Answer>A/B/C</Answer> otherwise your response will not be processed.

Table A8: Base prompt

For the nodes X and Y which form an edge in a Causal Graph, you have to identify which cause-and-effect relationship is more likely between the nodes of the edge. This will be used to rearrange the nodes in the edge to create a directed edge which accounts for causal relation from one node to another in the edge.

A. changing the state of node X causally affects a change in another node Y.

B. changing the state of node Y causally affects a change in another node X.

C. There is no causal relation between the nodes X and Y.

You can also take the edges from the skeleton which have been rearranged to create a directed edge to account for causal relationship between the nodes: directed_edges.
Make sure to first output a factually grounded reasoning for your answer. First give your reasoning and after that please make sure to provide your final answer within the tags <Answer>A/B/C</Answer>.
It is very important that you output your final answer between the tags like <Answer>A/B/C</Answer> otherwise your response will not be processed.

Table A9: Iterative orientation prompt

For the following undirected edge in a Causal Graph made of nodes X and Y, you have to identify which cause-and-effect relationship is more likely between the nodes of the edge. This will be used to rearrange the nodes in the edge to create a directed edge which accounts for causal relation from one node to another in the edge.

A. changing the state of node X causally effects a change in another node Y.

B. changing the state of node Y causally effects a change in another node X.

C. There is no causal relation between the nodes X and Y.

You can also take the other directed edges of nodes X: X_edges and Y: Y_edges of the Causal graph as context to redirect the edge to account for causal effect.
Make sure to first output a factually grounded reasoning for your answer. First give your reasoning and after that please make sure to provide your final answer within the tags <Answer>A/B/C</Answer>.
It is very important that you output your final answer between the tags like <Answer>A/B/C</Answer> otherwise your response will not be processed.

Table A10: Markov Blanket prompt

For the following edge in a given Undirected graph which is the skeleton of a Causal Graph: edge, you have to identify which cause-and-effect relationship is more likely between the nodes of the edge. This will be used to rearrange the nodes in the edge to create a directed edge which accounts for causal relation from one node to another in the edge.

A. changing the state of node X causally effects a change in another node Y.

B. changing the state of node Y causally effects a change in another node X.

C. There is no causal relation between the nodes X and Y.

You can take the whole skeleton of the causal graph as context for arriving at your answer, Undirected Graph: UG. Make sure to first output a factually grounded reasoning for your answer. First give your reasoning and after that please make sure to provide your final answer within the tags <Answer>A/B/C</Answer>.
It is very important that you output your final answer between the tags like <Answer>A/B/C</Answer> otherwise your response will not be processed.

Table A11: Providing Undirected graph as context

*Identify the causal relationships between the given variables and create a directed acyclic graph to context. Make sure to give a reasoning for your answer and then output the directed graph in the form of a list of tuples, where each tuple is a directed edge. The desired output should be in the following form: [('A','B'), ('B','C')] where first tuple represents a directed edge from Node 'A' to Node 'B', second tuple represents a directed edge from Node 'B' to Node 'C'and so on.*

*If a node should not form any causal relationship with other nodes, then you can add it as an isolated node of the graph by adding it seperately. For example, if 'C' should be an isolated node in a graph with nodes 'A', 'B', 'C', then the final DAG representation should be like [('A','B'), ('C')].*
*Use the description about the node provided with the nodes in brackets to form a better decision about the causal direction orientation between the nodes.*

*It is very important that you output the final Causal graph within the tags <Answer></Answer>otherwise your answer will not be processed.*

*Example:*
*Input: Nodes: ['A', 'B', 'C', 'D'];*
*Description of Nodes: [(description of Node A), (description of Node B), (description of Node C), (description of Node D)]*
*Output: <Answer>[('A','B'),('C','D')]</Answer>*
*Question:*
*Input: Nodes: nodes*
*Description of Nodes:*
*Output:*

Table A12: The structure of the triplet prompt.

Input: **('Right C7','Discoligment injury C6-C7')**

Answer: Discoligment injury C6-C7 can cause compression of the nerve roots that exit the spinal cord at the C7 level, which can lead to symptoms such as pain, numbness, and weakness in the right C7 dermatome. Therefore, the answer is <Answer>B</Answer>.

Input: **('Right C7', 'Left C7')**

Answer: Right C7 and left C7 are both parts of the cervical spine and are not known to directly influence each other. Therefore, the answer is <Answer>C</Answer>.

Input: **('Right elbow trouble', 'Left shoulder trouble')**

Answer: There is no direct causal relationship between right elbow trouble and left shoulder trouble. They may both be symptoms of a larger underlying condition, but they do not directly cause or affect each other. Therefore the answer is <Answer>C</Answer>.

Table A13: Example LLM (GPT-3.5-turbo) reasoning outputs for estimating causal directionality between different pairs of nodes using CoT Prompt (refer Table A3 for the prompt) for Neuropathic subgraph (used for pain diagnosis).