# OpenReview forum: "Causal Inference using LLM-Guided Discovery"
_AAAI.org/2024/Workshop/LLM-CP — LLM-CP @ AAAI 2024 Oral_

### Official Review · Reviewer_1L1P · 2023-12-04
**Accept**

**Rating:** 2
**Confidence:** 3

**Review:**

Relevant motivation, relevant references, relevant experiments

---

### Official Review · Reviewer_DGd2 · 2023-12-05
**Promising attempt for causal order using LLMs and discovery algorithms.**

**Rating:** 2
**Confidence:** 2

**Review:**

This work proposes the promising approach of prompting and voting strategy to get more reliable estimated causal graph, and further adopting discovery methods to obtain the final causal graph. The work also justifies causal order metric for effect estimation.

Strengths:
1. The proposed method achieves promising causal discovery compared to baseline discovery methods.
2. The proposed method introduces new idea of how to leverage LLM to generate better causal order.

Weaknesses:
1. LLM requires meaningful variable names.
2. How does LLM solve the ambiguitions in the variable names?
3. Comparison of the running time of different methods. How would the proposed method scale with the number of variables increasing?

Some other questions:
1. Is there any reason of the choice of GPT 3.5 and GPT 4?
2. The results shown in Table 1 are promising. How about also comparing with the baseline of simple pairwise variables prompt for LLMs or whole graph generation?
3. How about including more variables in each group of variables?

---

### Meta-Review · Area_Chair_cjXL · 2023-12-10

**Recommendation:** 2
**Confidence:** 3

**Metareview:**

The paper builds on work to use LLMs to generate causal graphs from background domain knowledge, but offers a promising strategy to get around the inherent context-dependence of causal edges, namely that it depends on which other variables are present. The paper is well written and the results look good, as is confirmed by both reviewers. Therefore I recommend accepting the paper for the workshop.

---

### Decision · Program_Chairs · 2023-12-14

**Decision:**

Accept (Oral)

**Comment:**

Thank you for submitting your work to the LLM-CP workshop @ AAAI 2024. See you in Vancouver!